# Health care utilization, mental disorders and behavioural disorders among perpetrators of intimate partner homicide in 2000–2016: A registry-based case-control study from Sweden

Solveig Lövestad[1,2☯]*, Karin Örmon[2,3‡], Viveka Enander[2,4‡], Gunilla Krantz[1,2☯]

1 Department of Public Health and Community Medicine, Institution of Medicine, University of Gothenburg, Gothenburg, Sweden, 2 Västra Götaland Region Competence Centre on Intimate Partner Violence (VKV), Gothenburg, Sweden, 3 Department of Health, Blekinge Institute of Technology, Karlskrona, Sweden, 4 Department of Social Work, Faculty of Social Science, Gothenburg University, Gothenburg, Sweden

☯ These authors contributed equally to this work.
‡ KO and VE also contributed equally to this work.
* Solveig.lovestad@gu.se

**Data Availability Statement:** This study is based on data available from the Swedish NBHW and Statistics Sweden. Data were used under license

## Abstract

Little is known about intimate partner homicide (IPH) perpetrator´s healthcare contacts and mental health problems before the killing. The aim was to compare male and female IPH perpetrators with matched controls from the general population by analysing differences in healthcare utilization and mental and behavioural disorders. This study includes 48 males and 10 females who perpetrated IPH between 2000 and 2016 in the Västra Götaland Region of Sweden. Controls (n = 458) were randomly selected from the general population and matched for sex, birth year and residential area. Data were retrieved from the Swedish National Patient Register and the Western Swedish Healthcare Register. Mental and behavioural disorders were classified according to ICD-10 (F00-F99). The Mann-Whitney U test was used to test for differences in health care utilization and mental and behavioural disorders. Compared to their controls, male perpetrators had more registered contacts with primary care ≤ 30 (p = < .001) and ≤ 365 days (p = .019), respectively, before the homicide; with specialist outpatient care ≤ 30 (p = < .001) and ≤ 365 days (p = < .001), respectively, before the homicide: and with inpatient care ≤ 30 (p = < .001) and ≤ 365 days (p = .024), respectively, before the homicide. Female perpetrators had more specialized outpatient care (p = .040) and inpatient care (p = .003) contacts ≤ 365 days before the homicide, compared to controls. Male perpetrators had at least one mental or behavioral disorder diagnosed in any studied healthcare setting except in inpatient care ≤ 30 days before homicide. Female perpetrators had more mental health disorders diagnosed in specialized outpatient care ≤ 365 days before the homicide (p < .001). Perpetrators had more healthcare contacts and mental disorders one year and one month prior to the homicide compared to their controls. Health care professionals should obtain necessary skills in routinely enquiring about intimate partner violence perpetration.

for the current study and cannot be shared publicly due to restrictions stipulated by the registry holders and Swedish Ethical Review Authority. Data are available from The Västra Götaland Region Competence Centre on Intimate Partner Violence (contact via Jenny Ström: jenny.strom@vgregion. se, Tel +46 768-40 22 96) upon reasonable request and with permission of the Swedish NBHW, Statistics Sweden and the Swedish Ethical Review Authority.

**Funding:** The author(s) received no specific funding for this work.

**Competing interests:** The authors have declared that no competing interests exist.

## Introduction

Intimate partner violence (IPV) is a major public health concern and may, in extreme cases, result in intimate partner homicide (IPH). IPH is defined as intentional violence that leads to death, i.e. manslaughter, involuntary manslaughter or murder perpetrated by a current or former intimate partner [1]. Apart from the premature death of the victim, IPH may lead to adverse, long-term consequences in the mental health and well-being of bereaved children, family members and bystanders [2, 3].

Previous research has shown that approximately 13.5% of all homicides worldwide are committed by a current or former intimate partner [4]. The vast majority of these victims are women, accounting for 39% of all female homicide victims [4]. For males, the corresponding percentage is much lower, accounting for 6% of all male homicide victims [4]. Recent data from Sweden are consistent with global trends; 52% (*n* = 13) of all female homicides and 4% (*n* = 4) of all male homicides were perpetrated by a former or current intimate partner [5]. Despite the introduction of new policies and increased awareness about IPH, the number of IPH victims in Sweden has remained relatively constant over the past three decades. Cases averaged 20 per year during the 1990´s and the beginning of the twenty-first century, followed by a decrease to an annual average of 16 cases between 2008 and 2017 [5]. A recently published report from Sweden showed that the average number of IPH victims had increased to 18 per year between 2018 and 2020 [6]. These findings demonstrate that further efforts are required to reduce the number of IPH victims and to obtain more knowledge about the perpetrators. This may lead to development of appropriate prevention methods and improve the possibility of identifying individuals at risk of perpetrating IPH.

It is well established that the risk of perpetrating IPH is influenced by multiple, intersecting determinants ranging from individual to societal level, with either buffering or exacerbating effects [7]. Earlier research has identified important factors associated with male- perpetrated IPH, such as previous and reiterated perpetration of IPV [8, 9], access to guns [8, 10], making previous threats with a weapon [8, 10], separation with feelings of abandonment [11], stalking and controlling behaviours [8, 10, 11]. Furthermore, if the male perpetrator of IPV has less than a high school education and/or is unemployed it tends to increase the likelihood of an IPH [8]. Regarding female- perpetrated IPH, research reveals that the perpetrator at the time of the homicide, commonly is unemployed and has a criminal history recorded by the police [12]. Most studies reveal that female perpetrated IPH tends to occur as an act of self-defence [13] and in response to being exposed to IPV [7, 14].

While previous research on risk factors for perpetrating IPH has contributed with valuable knowledge, there is still limited understanding about perpetrators' healthcare contacts before the killing. Although some research suggests that only a minority of those who perpetrate IPH are in contact with the healthcare system prior to the homicide [15], there is also evidence indicating that a considerable proportion of perpetrators do have contact with healthcare providers before the offence [16]. A previous study performed in the US, showed that approximately 20% of the IPH perpetrators had been in contact with a healthcare provider, concerning their physical or mental health or for substance abuse treatment, during the year prior to the homicide [9]. A study from Australia on male and female IPH perpetrators found that the most frequent type of service contact among perpetrators was the health care system, accounting for 54.2% of all service contacts [16]. Sharps and colleagues [9] found that among perpetrators who were reported to have fair or poor health, 15% had visited a healthcare provider for mental health problems, and almost 53% had visited a healthcare provider for physical health problems in the year prior to the IPH. Additionally, societal norms and expectations related to masculinities may influence on men´s ability to recognize their symptoms of distress as signs of mental health

problems [17]. It may be more socially acceptable for men to seek healthcare for physical health issues rather than for mental health problems [17]. Consequently, many men may primarily seek healthcare for reasons other than explicit mental health problems [17]. The majority of IPH perpetrators are men; thus, it is reasonable to assume that male IPH perpetrators may contact healthcare services for a range of physical problems rather than mental health issues. Thus, examining perpetrator´s general healthcare utilization regardless of diagnosis and healthcare service, may enhance our understanding of IPH perpetrators general health care utilization and well-being. While there is limited research on male perpetrators' healthcare utilization, much less is known about female perpetrators and their healthcare utilization before they commit the homicide. From previous research it is well established that women exposed to IPV are more likely to use health care services than women who have not experienced such violence [18, 19]. A study involving female perpetrators of IPH, revealed that approximately half of them had a history of exposure to IPV [16]. Among these female perpetrators who had experienced IPV exposure, the second most prevalent point of contact before perpetrating the homicide was with a healthcare provider [16]. Thus, very little is known about female perpetrators general healthcare contacts and even less is known if there are any differences in healthcare utilization between female perpetrators and women from the general population.

Existing research reveals that a history of mental health problems, such as depression [20–22], personality disorders [3, 21, 22] or suicidal ideation and attempts [23], is associated with male- perpetrated IPH. A recently published meta-analysis of male IPH perpetrators found that a history of mental health problems increased the likelihood of perpetrating IPH by 30% [8]. Furthermore, a study conducted in Sweden showed that 6.5% of all males perpetrating IPH between 2007 and 2009 had obtained inpatient and/or outpatient care from psychiatric services in the year prior to the homicide [24]. Mental health problems among female IPH perpetrators remains largely unexamined. Only a few studies have investigated mental health problems disaggregated by male and female IPH perpetrators [25] or focused exclusively on female perpetrators [12]. A study conducted in Sweden, revealed that two- thirds (n = 6) of the female perpetrators and one- third (n = 12) of the male perpetrators had previously been treated in psychiatric inpatient care [25]. Yet another study on female perpetrators of IPH, found that 42% had a diagnosed mental health condition at the time of the homicide, and 15% had previously experienced mental health issues [12]. The lack of studies regarding healthcare utilization and mental health issues among female perpetrators of IPH, clearly indicate that more research is needed on this topic. From previous research conducted in Sweden, we have obtained valuable knowledge about specific psychiatric diagnoses among IPH perpetrators [3, 21, 25]. However, investigating mental and behavioural disorders regardless of explicit diagnoses, may provide additional knowledge indicating if IPH perpetrators live with diagnosed mental illness to a higher extent than individuals from the general population.

To date, several studies have aimed at identifying factors associated with IPH perpetration by comparing IPH perpetrators with perpetrators of non-lethal IPV [23, 26] and perpetrators of family homicides, i.e. killing of family members other than an intimate partner [22]. Moreover, studies have compared male IPH perpetrators with male killers of female and male non-intimates, such as neighbours, strangers and acquaintances outside the family [21, 24, 27, 28]. A number of studies have aimed at ascertaining whether men who killed their intimate partners (IP) were similar to or more conventional than men who killed non-intimates, in terms of criminal history, background factors and mental health problems [27–29]. In their study comparing male IPH perpetrators with men who had killed other men, Dobash and colleagues [28] found that 27.5% of the IPH perpetrators and 24.7% of the men who had killed other men suffered from mental health problems during adulthood. Likewise, Loinaz and colleagues [29] showed in their study that there was no statistically significant difference, regarding mental

health problems, between men who killed female IPs and men who killed women outside an intimate relationship. However, Thomas et al. [27] found that a larger proportion of men who had killed their female or male IP had at some point in life been diagnosed with severe mental illness, i.e. a lifetime clinical diagnosis of major depression, bipolar disorder, psychotic disorder or mania, compared to men who had killed male or female non-intimates (11.6% and 25.6% respectively). Although the results are not conclusive, these studies provide important insights into the differences between perpetrators of IPH and perpetrators of other types of homicides. However, much less is known about the differences in mental health problems and healthcare utilization between perpetrators of IPH and the general population. To our knowledge, only one previous study, from Sweden and covering the period 1973–2009, has compared male IPH perpetrators with matched controls from the general population. In that study, Lysell et al. [3] found that 16.9% of the perpetrators and 4.2% of the matched controls had been hospitalised for any mental disorder. However, previous research has not yet examined the differences between female IPH perpetrators and women from the general population. Examining potential differences between male and female IPH perpetrators and individuals from the general population is important for future public health interventions and guideline recommendations. Although previous research is not conclusive, social and mental health issues may be less prevalent among perpetrators of IPH compared to perpetrators of other types of homicides and therefore, IPH perpetrators may seem to be more "conventional" and less likely to be socially disadvantaged compared to perpetrators of other types of homicide [28, 30]. However, comparing IPH perpetrators with individuals from the general population without any confirmed history of IPH perpetration, may provide a more accurate identification of the magnitude, as well as the patterns of healthcare utilization and mental and behavioural disorders among perpetrators of IPH. Furthermore, comparing sociodemographic factors among IPH perpetrators with healthcare contacts prior to the homicide with the same factors among individuals from the general population with healthcare contacts may contribute to improving identification of individuals at risk of perpetrating IPH. This study seeks to fill the knowledge gap by identifying differences between male and female IPH perpetrators and males and females from the general population. Thus, the aims of this case-control study were to:

1. Compare healthcare contacts, in primary care, specialized outpatient care and inpatient care, between male and female IPH perpetrators and their matched controls in the month ($\leq$ 30 days) and in the year ($\leq$ 365 days) prior to the homicide (both studied intervals include the day of the homicide)

2. Compare mental and behavioural disorders (ICD-10), diagnosed in primary care, specialized outpatient care or inpatient care, between male and female IPH perpetrators and their matched controls in the month ($\leq$ 30 days) and in the year ($\leq$ 365 days) prior to the homicide (both studied intervals include the day of the homicide)

3. Investigate whether the distribution of sociodemographic factors, i.e. employment status, educational level and receiving social benefits, differed between IPH perpetrators and matched controls with registered healthcare contacts

## Materials and methods

### Study design and study population

In this study, an IP is defined as a current or former spouse, fiancé/e, cohabiting partner or boy-/girlfriend, irrespective of sexual identity. IPH includes acts of violence leading to the death of an IP, i.e. manslaughter, involuntary manslaughter and murder.

This study is a registry-based, case-control study including all identified cases of male- and female- perpetrated IPH between January 1, 2000 and December 31, 2016 in the Västra Götaland Region (VGR) of Sweden. VGR has a population of 1.7 million and consists of both rural and urban areas, including Gothenburg, the region's largest city and the second largest city in Sweden. Gothenburg has an inner-city population of around 560 000 [31]. This study is part of a larger research project called the IPH-Stop study, including all IPH cases (perpetrators and victims) in VGR during 2000–2016. The design of the IPH-Stop study as well as the characteristics of all included IPH cases are presented in more detail elsewhere [14, 32].

After ethical approval was granted, IPH perpetrators were identified through police records and preliminary enquiries kept at the regional police authority. District court records were obtained for additional verification. Linkage to national and regional registries was achieved with the unique personal identification number (PIN) assigned to all residents at birth or immigration and used across all national registries [33]. For each perpetrator, Statistics Sweden which is a governmental agency [33], randomly selected 10 general population controls from the Swedish Total Population Register, matched for sex, birth year and residential area at the time of the IPH. This approach, i.e., the random selection of general population controls matched at a ratio of 10:1, has been employed in several registry-based studies conducted in Sweden [3, 34–37]. The inclusion criterion was minimum age 18 years. The total sample consisted of 638 individuals: 48 male and 10 female perpetrators and their matched controls. The perpetrators had killed a current or previous opposite-sex partner, except for one case in which a male perpetrator committed the homicide within a same-sex relationship.

## The Swedish health care system

The Swedish healthcare system is primarily tax-funded. Providers are either public or publicly funded private units and patient fees constitute only a small amount of total health care funding [38]. The healthcare system is divided into primary healthcare, specialized outpatient care and inpatient care. Primary healthcare, staffed by general practitioners, district nurses, physiotherapists, psychologists and counsellors, is generally the first provider that the patient visits and offers treatment of the most common conditions and illnesses [38]. Specialized outpatient care [38] is a hospital unit or consists of an independent outpatient clinic [39]. Inpatient care is hospital care, including by referral from other providers or after transfer from an accident and emergency ward [39].

## Data collection

Data on primary care contacts were retrieved from the Western Swedish Healthcare Register (VEGA), which covers all healthcare contacts provided by public and private providers in VGR since 2000. Information on inpatient and specialized outpatient care contacts was collected from The Swedish National Patient Register (NPR). The NPR is administrated by the National Board of Health and Welfare (NBHW) and covers all in-patient care in Sweden since 1987. Beginning in January 2001, it also covers outpatient visits to specialist services, including day surgery and psychiatric care, provided by both private and public caregivers [40, 41]. Primary care contacts are not covered by the NPR. As the NPR does not cover outpatient visits before 2001, we lack data on outpatient specialist care $\leq$ 30 and $\leq$ 365 days before index (i.e. before the killings committed between January 2000 and January 2001) for three perpetrators and their matched controls.

Sociodemographic variables, i.e. the highest completed level of education, employment status and receiving social benefits, were retrieved from the longitudinal

integration database for health insurance and labour market studies (LISA) held by Statistics Sweden. Data in LISA are linked individually to different population registries and are available from 1990 and onwards [42].

## Variables

**Health care utilization.** Data on health care utilization were extracted from VEGA and NPR. Individuals with at least one registered contact with a healthcare provider during the study period 2000–2016 were included. Health care utilization was defined as the number of *recorded diagnosis of any kind* in primary care (VEGA), inpatient care and/or specialized outpatient care (NPR) during the month ($\leq$ 30 days) and during the year ($\leq$ 365 days) respectively, before the homicide. If an individual had more than one recorded diagnosis per date in any of the three health care settings, only one diagnosis per date and healthcare setting was included for further analysis. For the purpose of descriptive analysis, primary care, specialized outpatient care and inpatient care utilization was categorized as at least one recorded contact in the month prior to the homicide (= 1) or as no recorded contact in the month prior to the homicide (= 0). The same applied to the different healthcare settings for the year prior to the offence. The time frame one month prior to the homicide thus overlaps with the time frame one year prior to the homicide and both intervals include the day of the homicide.

**Mental and behavioural disorders.** Data on diagnosed mental and behavioural disorders were retrieved from VEGA and NPR. Perpetrators and controls with at least one diagnosis according to the ICD-10 classification of mental and behavioural disorders (F00-F99) in the month ($\leq$ 30 days) and the year ($\leq$ 365 days) prior to the homicide were included. If an individual had more than one mental or behavioural disorder diagnosis (F00-F99) per date and healthcare setting, only one diagnosis per date and setting was included for further analysis. For the purpose of descriptive analysis, mental and behavioural disorders diagnosed in primary care, specialized outpatient care and inpatient care were categorized as at least one recorded diagnosis in the month prior to the homicide (= 1) or no diagnosis in the month prior to the homicide (= 0). The same applied to mental and behavioural disorders in the different healthcare settings in the year prior to the offence.

**Sociodemographic variables.** In order to ascertain whether IPH perpetrators with recorded healthcare contacts differed from their matched controls with recorded healthcare contacts in terms of employment status, highest achieved educational level and receiving social benefits, these variables were extracted for the year preceding the crime for both perpetrators and controls. *Employment* was categorized as gainfully employed or no employment. *Receiving social benefits, i.e. receiving economical support and financial assistance for living expenses*, was categorized as having received social benefits during the year prior to index (= 1) or not (= 0). *Educational level* was categorized as 'pre-secondary education ($\leq$9 years)', 'secondary education (9–12 years)' and 'post-secondary education ($>$ 12 years)'. Due to small cell frequencies and for additional tests for difference, the variable was further transformed into 'post-secondary education' and 'pre-/ secondary education'.

**Statistical analysis.** Analyses were carried out using the Statistical Package for the Social Sciences (SPSS), version 25. Descriptive statistics on sociodemographic characteristics, healthcare utilization and mental and behavioural disorders were presented as frequencies (n) and percentages (%). In order to test for differences in healthcare utilization and mental and behavioural disorders between male and female IPH perpetrators and their matched controls, the non-parametric Mann-Whitney U test ($p < .05$) was used for non-normally distributed continuous variables. Results based on Mann-Whitney U test were presented with mean ranks. Analyses of healthcare utilization and mental and behavioural disorders within 30 and 365

days, respectively, before the homicide were performed separately for male and female perpetrators and controls.

When testing for differences in the distribution of socio-demographic characteristics between IPH perpetrators and controls with registered healthcare utilization in the year prior to the offence, the chi-squared test was used with the Fisher's Exact probability test (p < .05) for categorical variables with expected frequency less than five. Due to small cell frequencies, tests for differences in sociodemographic characteristics related to healthcare contacts ≤ 30 days before the homicide and inpatient care contacts were not applicable. Analyses were performed with female and male perpetrators as a unitary construct. According to recommendations by Green [43] regarding small sample size, no further multivariable analyses were performed.

**Ethical considerations.**   Ethical approval was provided by the Regional Ethical Review Board in Gothenburg (approval number Dnr: 434–16). All data were collected by the NBW and Statistics Sweden. The data was anonymised by SCB by replacing the subjects PIN´s with serial numbers before the data was sent to the researchers. In accordance with the Regional Ethical Review Board in Gothenburg, informed consent is not required for registry-based research. The registries used for this study are open to researchers upon request if ethical approval has been provided and the request is deemed appropriate by the authority maintaining the registry.

## Results

### Sociodemographic characteristics

The sample consisted of 48 male perpetrators and their matched controls (n = 480), with an average age of 47.5 years (standard deviation = 16.18) at the time of index, as well as 10 female perpetrators and their matched controls (n = 100), with an average age of 42.8 years (standard deviation = 11.10) at the time of index. The proportion of male perpetrators (82.8%) was higher than that of female perpetrators (17.2%). Of the 48 male perpetrators, eight committed suicide soon after the homicide. A larger proportion of male and female perpetrators (47.9% vs 70.0%) were unemployed, compared to their matched controls (20.6% and 23.0% respectively) (Table 1). Furthermore, 12.5% of the male perpetrators and none of the female perpetrators

**Table 1. Demographic characteristics of male and female perpetrators of intimate partner homicide (IPH) and their matched controls.**

|  | Males (N = 528) | | Females (N = 110) | |
|---|---|---|---|---|
|  | Perpetrators Total N = 48 | Controls Total N = 480 | Perpetrators Total N = 10 | Controls Total N = 100 |
|  | n (%) | n (%) | n (%) | n (%) |
| **Suicide** | 8 (16.7) | 80 (16.7) | 0 (0.0) | 0 (0.0) |
| **Employment status** |  |  |  |  |
| Employed | 16 (33.3) | 300 (62.5) | 3 (30.0) | 76 (76.0) |
| Unemployed | 23 (47.9) | 99 (20.6) | 7 (70.0) | 23 (23.0) |
| Missing | *9 (18.8)* | *81 (16.9)* | *0 (0.0)* | *1 (1.0)* |
| **Education** |  |  |  |  |
| Pre-secondary education, ≤ 9 years | 9 (18.8) | 93 (19.4) | 3 (30.0) | 12 (12.0) |
| Secondary education, 9–12 years | 28 (58.3) | 223 (46.5) | 6 (60.0) | 48 (48.0) |
| Post-secondary education, > 12 years | 6 (12.5) | 128 (26.7) | 0 (0.0) | 38 (38.0) |
| Missing | *5 (10.4)* | *36 (7.5)* | *1 (10.0)* | *2 (2.0)* |
| **Receiving social benefits** |  |  |  |  |
| Yes | 10 (20.8) | 23 (4.8) | 2 (20.0) | 3 (3.0) |
| No | 38 (79.2) | 457 (95.2) | 8 (80.0) | 97 (97.0) |

had a post- secondary education, whereas the corresponding respective proportions for male and female controls were 26.7% and 38.0%. A higher proportion of male and female perpetrators were receiving benefits (20.8% and 20.0%, respectively) compared to their matched controls (4.8% and 3.0%, respectively).

## Frequency of healthcare contacts within 30 and 365 days before the homicide

Male perpetrators (mean rank = 251.3) had significantly more primary care contacts (22.6%) within 30 days before the homicide than controls (4.8%) (mean rank = 212.7) ($U$ = 7293.5, $p < .001$) (Table 2). Male perpetrators (mean rank = 251.3) also had more primary care contacts within 365 days before the homicide than controls (mean rank = 212.7) (35.5% and

**Table 2. Recorded healthcare contacts among male perpetrators (n = 48) of Intimate partner homicide and their matched controls (n = 480) in Sweden 2000–2016.**

| Recorded healthcare contacts | | Male perpetrators | | Male controls | | | |
|---|---|---|---|---|---|---|---|
| **Primary care contacts ≤ 30 days before homicide**[a] *(Perpetrators N = 31/Controls N = 399)* | | n (valid %) | Mean Rank | n (valid %) | Mean Rank | U | P-value |
| Valid | No contact | 24 (77.4) | | 380 (95.2) | | | |
| | ≥ 1 contact [max number of contacts] | 7 (22.6) [6] | 251.3 | 19 (4.8) [9] | 212.7 | 7293.5 | < .001* |
| Missing | Not registered | 17 | | 81 | | | |
| **Primary care contacts ≤ 365 days before homicide**[a] *(Perpetrators N = 31/Controls N = 399)* | | | | | | | |
| Valid | No contact | 20 (64.5) | | 320 (80.2) | | | |
| | ≥ 1 contact [max number of contacts] | 11 (35.5) [33] | 251.3 | 79 (19.8) [115] | 212.7 | 7293.0 | **.019*** |
| Missing | Not registered | 17 | | 81 | | | |
| **Specialized outpatient care contacts ≤ 30 days before homicide**[a] *(Perpetrators N = 44/Controls N = 439)* | | | | | | | |
| Valid | No contact | 34 (77.3) | | 422 (96.1) | | | |
| | ≥ 1 contact [max number of contacts] | 10 (22.7) [2] | 284.0 | 17 (3.9) [1] | 237.8 | 11504.0 | < .001* |
| Missing | Index occurred before register was established. | 3 | | 30 | | | |
| | Not registered | 1 | | 11 | | | |
| **Specialized outpatient care contacts ≤ 365 days before homicide**[a] *(Perpetrators N = 44/ Controls N = 439)* | | | | | | | |
| Valid | No contact | 18 (40.9) | | 326 (74.3) | | | |
| | ≥ 1 contact [max number of contacts] | 26 (59.1) [7] | 315.9 | 113 (25.7) [16] | 234.6 | 12909.0 | < .001* |
| Missing | Index occurred before register was established. | 3 | | 30 | | | |
| | Not registered | 1 | | 11 | | | |
| **Inpatient care contacts ≤ 30 days before homicide**[a] *(Perpetrators N = 43, Controls N = 354)* | | | | | | | |
| Valid | No contact | 36 (83.7) | | 346 (97.7) | | | |
| | ≥ 1 contact [max number of contacts] | 7 (16.3) [1] | 223.8 | 8 (2.3) [1] | 196.0 | 8678.0 | < .001* |
| Missing | Not registered | 5 | | 126 | | | |
| **Inpatient care contacts ≤ 365 days before homicide**[a] *(Perpetrators N = 43, Controls N = 354)* | | | | | | | |
| Valid | No contact | 34 (79.1) | | 320 (90.4) | | | |
| | ≥ 1 contact [max number of contacts] | 9 (20.9) [4] | 196.6 | 34 (9.6) [5] | 219.1 | 8473.0 | **.024*** |
| Missing | Not registered | 5 | | 126 | | | |

[a] Including the day of the homicide

*P < .05

19.8% respectively) (U = 7293.0, $p$ = .019). A higher proportion of male perpetrators (22.7%) (mean rank = 284.0) than controls (3.9%) (mean rank = 237.8) had at least one specialized outpatient care contact in the 30 days before the homicide (U = 11504.0, $p$ < .001). Furthermore, the number of specialized outpatient care contacts in the 365 days before the homicide was considerably higher (59.1%) among male perpetrators (mean rank = 315.9), compared to their matched controls (25.7%) (mean rank 234.6), (U = 12909.05, $p$ < .001). About 16.3% of the male perpetrators (mean rank = 223.8) had at least one inpatient care contact within 30 days prior to the homicide, compared with 2.3% of the matched controls (mean rank = 196.0), (U = 8678.0, p < .001). Moreover, male perpetrators (mean rank 196.6) had more inpatient care contacts in the 365 days before the homicide (20,9%) compared to 9.6% of the controls (mean rank = 219.1), (U = 8473.0, $p$ = .024).

Among women, statistically significant differences were found in the number of specialized outpatient care contacts within 365 days before the homicide. Female perpetrators (mean rank = 68.2) had more registered care contacts (60.0%) than their matched controls (mean rank = 51.4) (26.3%) (U = 626.5, p = .040) (Table 3). Furthermore, female perpetrators (mean rank = 69.1) had a higher number of inpatient care contacts in the 365 days before the homicide compared to controls (mean rank = 50.4) (44.4% and 11.7% respectively) (U = 576.5, p = .003).

## Diagnosed mental and behavioural disorders within 30 and 365 days before the homicide

Male perpetrators had more mental or behavioural disorders diagnosed in any primary care setting both within 30 days and within 365 days before the homicide, compared to their matched controls (p < .001) (Table 4). Among the men 16.1% of the IPH perpetrators (mean rank = 245.6) and 1.0% of their matched controls (mean rank = 213.2) had at least one mental or behavioural disorder diagnosed in primary care during the month prior to the homicide (U = 7117.0, $p$ < .001). Of the male perpetrators, 9.1% had at least one mental or behavioural disorder diagnosed in the specialized outpatient special care setting in the month prior to the homicide, compared to 0.7% of the control group (mean rank = 260.4 and 240.2, respectively, U = 10470.0, $p$ = .001). Furthermore, more male perpetrators (mean rank = 289.6) (25.0%) had at least one mental or behavioural disorder diagnosed in outpatient care during the 365 days prior to the homicide than matched controls (mean rank = 237.2) (3.4%) (U = 11754.0, $p$ < .001). Among males, 7.0% of the perpetrators (mean rank = 207.9) and 2.0% of controls (mean rank = 197.9) had at least one mental health disorder diagnosed in inpatient care during the year prior to the homicide (U = 7993.5, $p$ = .047).

Table 5 shows that female perpetrators (mean rank = 66.1) more frequently (30.0%) had at least one mental health disorder diagnosed in specialized outpatient care during the year prior to the killing than their matched controls (mean rank = 51.6) (2.1%) (U = 606, $p$ < .001).

## Sociodemographic characteristics related to healthcare contacts during the year prior to the homicide

A larger proportion (90.0%) of unemployed male and female perpetrators had contacted primary care during the year before the offence, compared to their unemployed matched controls (28.6%) (p < .001) (Table 6). Among perpetrators who received social benefits, 33.3% had been seen by a primary care provider during the year prior to the homicide, whereas the corresponding figure for their matched controls receiving social benefits was 5.8% (p = .012). Perpetrators who were unemployed utilized specialized outpatient care to a higher extent than their unemployed matched controls during the year prior to the homicide (61.3% vs 34.8%;

**Table 3. Recorded healthcare contacts among female perpetrators (n = 10) of Intimate partner homicide and their matched controls (n = 100) in Sweden 2000–2016.**

| Recorded healthcare contacts | | Female perpetrators | | Female controls | | | |
|---|---|---|---|---|---|---|---|
| **Primary care contacts 30 days before homicide[a]** *(Perpetrators N = 7/ Controls N = 86)* | | n (valid %) | Mean Rank | n (valid %) | Mean Rank | U | p-value |
| Valid | No contact | 7 (100.0) | | 84 (97.7) | | | |
| | ≥ 1 Registered contact [max. number of contacts] | 0 (0.0) | NA | 2 (2.3) [2] | NA | NA | NA |
| Missing | Not registered | 3 | | 14 | | | |
| **Primary care contacts 365 days before homicide[a]** *(Perpetrators N = 7/ Controls N = 86)* | | | | | | | |
| Valid | No contact | 6 (85.7) | | 79 (91.9) | | | |
| | ≥ 1 Registered contact [max. number of contacts] | 1 (14.3) [5] | NA | 7 (8.1) [4] | NA | NA | NA |
| Missing | Not registered | 3 | | 14 | | | |
| **Specialized outpatient care contacts 30 days before homicide[a]** *(Perpetrators N = 10/Controls N = 95)* | | | | | | | |
| Valid | No contact | 9 (90.0) | | 89 (93.7) | | | |
| | ≥ 1 Registered contact [max. number of contacts] | 1 (10.0) [2] | NA | 6 (6.3) [2] | NA | NA | NA |
| Missing | Not registered | 0 | | 5 | | | |
| **Specialized outpatient care contacts 365 days before homicide[a]** *(Perpetrators N = 10/Controls N = 95)* | | | | | | | |
| Valid | No contact | 4 (40.0) | | 70 (73.7) | | | |
| | ≥ 1 Registered contact [max. number of contacts] | 6 (60.0) [10] | 68.2 | 25 (26.3) [18] | 51.4 | 626.5 | **.040***  |
| Missing | Not registered | 0 | | 5 | | | |
| **Inpatient care contact 30 days before homicide[a]** *(Perpetrators N = 9, Controls N = 94)* | | | | | | | |
| Valid | No contact | 9 (100) | | 93 (98.9) | | | |
| | ≥ 1 Registered contact [max. number of contacts] | 0 (0.0) | NA | 1 (1.1) [1] | NA | NA | NA |
| Missing | Not registered | 1 | | 6 | | | |
| **Inpatient care contact 365 days before homicide** *(Perpetrators N = 9, Controls N = 94)* | | | | | | | |
| Valid | No contact | 5 (55.6) | | 83 (88.3) | | | |
| | ≥ 1 Registered contact [max. number of contacts] | 4 (44.4) [8] | 69.1 | 11 (11.7) [3] | 50.4 | 576.5 | **.003***  |
| Missing | Not registered | 1 | | 6 | | | |

[a] Including the day of the homicide

*P < .05

p = .007). Furthermore, 18.8% of the perpetrators who were receiving social benefits had recorded contacts with specialized outpatient care during the year prior to the killing; the corresponding figure for controls receiving social benefits was 5.1% (p = .018).

## Discussion

To our knowledge, this is the first study comparing the frequency of health care contacts and mental and behavioural disorders in male and female IPH perpetrators and their matched general population controls.

### Healthcare contacts during the month and year prior to the homicide

Overall, male IPH perpetrators had significantly more recorded contacts with all studied types of healthcare than their matched controls in the month and in the year before the homicide,

**Table 4. Diagnosed mental and behavioural disorders among male perpetrators (n = 48) of Intimate partner homicide and their matched controls (n = 480) in Sweden, 2000–2016.**

| Diagnoses according to ICD10 (F00-F99) | | Male perpetrators | | Male controls | | | |
|---|---|---|---|---|---|---|---|
| **In primary care ≤ 30 days before homicide**[a] *(Perpetrators N = 31/Controls N = 399)* | | n (valid %) | Mean Rank | n (valid %) | Mean Rank | U | p-value |
| Valid | No diagnosis | 26 (83.9) | | 395 (99.0) | | | |
| | ≥ 1 diagnosis [max number of diagnoses] | 5 (16.1) [2] | 245.6 | 4 (1.0) [3] | 213.2 | 7117.0 | < .001* |
| Missing | Not registered | 17 | | 81 | | | |
| **In primary care ≤ 365 days before homicide**[a] *(Perpetrators N = 31/Controls N = 399)* | | | | | | | |
| Valid | No diagnosis | 25 (80.6) | | 381 (95.5) | | | |
| | ≥ 1 diagnosis [max number of diagnoses] | 6 (19.4) [8] | 245.6 | 18 (4.5) [8] | 213.2 | 7115.0 | < .001* |
| Missing | Not registered | 17 | | 81 | | | |
| **In specialized outpatient care ≤ 30 days before homicide**[a] *(Perpetrators N = 44/Controls N = 439)* | | | | | | | |
| Valid | No diagnosis | 40 (90.9) | | 436 (99.3) | | | |
| | ≥ 1 diagnosis [max number of diagnoses] | 4 (9.1) [1] | 260.4 | 3 (0.7) [1] | 240.2 | 10470.0 | < .001* |
| Missing | Not registered | 4 | | 41 | | | |
| **In specialized outpatient care ≤ 365 days before homicide**[a] *(Perpetrators N = 44/Controls N = 439)* | | | | | | | |
| Valid | No diagnosis | 33 (75.0) | | 424 (96.6) | | | |
| | ≥ 1 diagnosis [max number of diagnoses] | 11 (25.0) [6] | 289.6 | 15 (3.4) [9] | 237.2 | 11754.0 | < .001* |
| Missing | Not registered | 4 | | 41 | | | |
| **In inpatient care ≤ 30 days before homicide**[a] *(Perpetrators N = 43/Controls N = 354)* | | | | | | | |
| Valid | No diagnosis | 40 (93.0) | | 354 (100) | | | |
| | ≥ 1 diagnosis [max number of diagnosis] | 3 (7.0) [1] | NA | 0 (0.0) | NA | NA | NA |
| Missing | Not registered | 5 | | 126 | | | |
| **In inpatient care ≤ 365 days before homicide**[a] *(Perpetrators N = 43/Controls N = 354)* | | | | | | | |
| Valid | No diagnosis | 40 (93.0) | 207.9 | 347 (98.0) | 197.9 | 7993.5 | .047* |
| | ≥ 1 diagnosis [max number of diagnosis] | 3 (7.0) [2] | | 7 (2.0) [2] | | | |
| Missing | Not registered | 5 | | 126 | | | |

[a] Including the day of the homicide

*P < .05

including the day of the homicide. This is in line with previous research indicating that healthcare services may be frequently utilized by perpetrators of IPV and IPH for a range of physical and mental health problems [16, 44]. Direct comparison of studies is difficult due to differences in healthcare systems and in ways of measuring healthcare contacts among IPH perpetrators. However, the proportions of healthcare contacts in the year prior to the killing found in this study, do corroborate previous research findings. For instance, we found that almost 21% of the male perpetrators with registered healthcare contacts during 2000–2016 had received inpatient care during the year prior to the killing. This concurs with the findings of a study by Sharps and colleagues [9], in which approximately 20% of the male perpetrators had been seen by a healthcare provider in the year prior to the killing. Furthermore, we found that of the 31 male perpetrators registered with primary care contacts, 35.5% had been seen by a primary healthcare provider during the year prior to the homicide. This is a somewhat lower proportion compared to the findings in the study by Murphy et al [16], in which 52.6% of the

**Table 5. Recorded mental and behavioural disorders among female perpetrators (n = 10) of Intimate partner homicide and their matched controls (n = 100) in Sweden, 2000–2016.**

| Diagnoses according to ICD10 (F00-F99) | | Perpetrators | | Controls | | | |
|---|---|---|---|---|---|---|---|
| **In primary care 30 days before homicide[a]** *(Perpetrators N = 7/Controls N = 86)* | | n (valid %) | Mean Rank | n (valid %) | Mean Rank | U | p-value |
| Valid | No diagnosis | 7 (100) | | 86 (100) | | | |
| | ≥ 1 diagnosis [max. number of diagnosis] | 0 (0.0) | NA | 0 (0.0) | NA | NA | NA |
| Missing | Not registered | 3 | | 14 | | | |
| **In primary care 365 days before homicide[a]** *(Perpetrators N = 7/Controls N = 86)* | | | | | | | |
| Valid | No diagnosis | 6 (85.7) | | 83 (96.5) | | | |
| | ≥ 1 diagnosis [max. number of diagnosis] | 1 (14.3) [2] | NA | 3 (3.5) [1] | NA | NA | NA |
| Missing | Not registered | 3 | | 14 | | | |
| **In special outpatient care 30 days before homicide[a]** *(Perpetrators N = 10/Controls N = 95)* | | | | | | | |
| Valid | No diagnosis | 10 (100.0) | | 95 (100.0) | | | |
| | ≥ 1 diagnosis [max. number of diagnosis] | 0 (0.0) | NA | 0 (0.0) | NA | NA | NA |
| Missing | Not registered | 0 | | 5 | | | |
| **In special outpatient care 365 days before homicide** *(Perpetrators N = 10/Controls N = 95)* | | | | | | | |
| Valid | No diagnosis | 7 (70.0) | | 93 (97.9) | | | |
| | ≥ 1 diagnosis [max. number of diagnosis] | 3 (30.0) [5] | 66.1 | 2 (2.1) [5] | 51.6 | 606 | < .001* |
| Missing | Not registered | 0 | | 5 | | | |
| **In inpatient care contact 30 days before homicide[a]** *(Perpetrators N = 9/Controls N = 94)* | | | | | | | |
| Valid | No diagnosis | 9 (100.0) | | 94 (100.0) | | | |
| | ≥ 1 diagnosis [max. number of diagnosis] | 0 (0.0) | NA | 0 (0.0) | NA | NA | NA |
| Missing | Not registered | 1 | | 6 | | | |
| **In inpatient care contact 365 days before[a] homicide** *(Perpetrators N = 9/Controls N = 94)* | | | | | | | |
| Valid | No diagnosis | 6 (66.7) | | 93 (98.9) | | | |
| | ≥ 1 diagnosis [max. number of diagnosis] | 3 (33.3) | NA | 1 (1.1) | NA | NA | NA |
| Missing | Not registered | 1 | | 6 | | | |

[a] Including the day of the homicide

*P < .05

male and female perpetrators had seen a general practitioner within 12 months prior to the killing. The prevention of IPH requires the early detection of exposure to and perpetration of IPV so that future acts of violence can be prevented. Both victims and perpetrators of IPV are likely to seek care [45–47] thus healthcare providers have an important role in the early identification of IPV. Since the primary care service generally is the first provider that the patient visits and they meet a wide range of patients [38, 48], our findings do indicate that healthcare providers in general and primary care providers in particular, have an important role in detecting individuals at risk of perpetrating IPH. The fact that about 59% of the perpetrators with specialized outpatient care contacts had at least one registered contact with this healthcare provider in the year before the homicide, indicates that this is another important healthcare setting for identifying individuals at risk of committing IPH.

In this study, female IPH perpetrators had more recorded specialized outpatient care contacts and inpatient care contacts within 365 days before the homicide, compared to their

**Table 6. Distribution of sociodemographic characteristics among male and female perpetrators of intimate partner homicide and their matched controls, with registered healthcare contacts ≤ 365 days before homicide.**

| Sociodemographic characteristics | Primary care contact ≤ 365 days before homicide** | | | Specialized outpatient care contact ≤ 365 days before homicide** | | |
|---|---|---|---|---|---|---|
| | Perpetrators n = 12 | Controls n = 86 | Chi²-test [a] | Perpetrators n = 36 | Controls n = 138 | Chi²-test [a] |
| | n (%) | n (%) | | n (%) | n (%) | |
| **Employment status** | | | | | | |
| Employed | 1 (9.1) | 60 (71.4) | < .001* | 12 (38.7) | 88 (65.2) | .007* |
| Unemployed | 10 (90.9) | 24 (28.6) | | 19 (61.3) | 47 (34.8) | |
| *Missing* [b] | *1* | *2* | | *5* | *3* | |
| **Education** | | | | | | |
| Post-secondary education | 0 (0.0) | 23 (27.1) | .061 | 3 (9.7) | 31 (23.3) | .092 |
| Pre-/Secondary education | 11 (100) | 62 (72.9) | | 28 (90.3) | 102 (76.7) | |
| *Missing* [b] | *1* | *1* | | *5* | *5* | |
| **Receiving social benefits** | | | | | | |
| No | 8 (66.7) | 81 (94.2) | .012* | 26 (81.3) | 131 (94.9) | .018* |
| Yes | 4 (33.3) | 5 (5.8) | | 6 (18.8) | 7 (5.1) | |
| *Missing* [b] | *0* | *0* | | *4* | *0* | |

[a] Fisher's exact text for categorical variables with expected frequency < 5

[b] Missing values were excluded from the Chi² analysis.

*P < .05

**including the day of the homicide

matched controls. These results further support the idea that IPH perpetrators seem to utilize healthcare services to a higher extent than men and women in the general population.

Studies indicate that screening or routine inquiry, i.e., inquiring about IPV in certain healthcare settings or when indicators of IPV are present, may increase the identification of IPV [47–49]. This is particularly true when questions about IPV are asked in a non-judgemental manner [48] and on repeated occasions [50], especially on follow-up visits [46]. Previous research reveals that when patients are asked about the use of IPV, they do not feel offended or react defensively if they feel comfortable and trust the healthcare provider asking about IPV [48]. Furthermore, it is important to address IPV as a healthcare issue, expressing concerns for the health and well-being of both the perpetrator, the victim, and their children [47]. In addition to this, clear communication and procedures following inquiries about IPV, such as documentation in electronic medical records, mandated reporting requirements, and the provision of clear referral services and/or future treatment options, have been stressed as important facilitators for disclosing IPV perpetration [48]. In line with this, healthcare services in Sweden are recommended, in addition to providing treatment, to refer patients in need of further support to the appropriate institutions and organizations with the right competencies and resources for addressing IPV [50]. This includes referral to social services and in some occasions, such as increased risk for IPH, it may also involve contacting the police [50].

## Diagnosed mental and behavioural disorders in the month and year prior to the homicide

We found that, compared to controls, male IPH perpetrators had significantly more mental and behavioural disorders diagnosed in primary care, specialized outpatient care and inpatient

care regardless of time frame, but this was not the case when it came to inpatient care ≤ 30 days prior to the offence. In this study, 16.1% of the male perpetrators had a mental and/or behavioural disorder diagnosed in primary care one month prior to the homicide. This is consistent with a population- based study from England and Wales, in which the authors found that 20% of male and female IPH perpetrators had symptoms of mental illness at the time of the homicide [22]. The only statistically significant finding among female perpetrators was that they had more mental and behavioural disorders diagnosed in specialized outpatient care during the year prior to the homicide, compared to controls. Only a few previous studies have examined healthcare contacts and mental health problems among female perpetrators [16, 22, 25]. These studies have either compared female IPH perpetrators with male perpetrators [25] or presented mental health problem results among female and male IPH perpetrators as a composite outcome [16, 22]. This makes direct comparison of our findings with those of previous research difficult. However, our findings indicate that female IPH perpetrators may suffer from mental and behavioural disorders to a higher extent than women in the general population. These findings must nonetheless be interpreted with caution, since the sample size is small. However, it is important to emphasize that our results do suggest that healthcare providers may play an important role in the identification and referral of individuals at risk of perpetrating IPH [16]. A Swedish fatality review published in 2022, which involved perpetrators of IPH and other types of homicide perpetrators with prior healthcare contacts, identified gaps in the healthcare system's ability to prevent subsequent homicides [6]. This encompassed cases where perpetrators had well-known risk factors such as a history of violence perpetration, as well as severe anxiety and/or substance abuse [6]. In these cases, healthcare providers either failed to inquire about violence and/or did not provide sufficient treatment or assistance to prevent further violence and homicide [6]. The review further revealed that even when some perpetrators disclosed severe anxiety, controlling behaviour and aggression following the separation from their IP´s, healthcare services failed to inquire about potential violence [6]. Based on the fatality review [6] and previous research linking mental health problems to IPH perpetration [20–23], we suggest that healthcare providers, but mental health services in particular, pay specific attention to patients who show or disclose aggressive and controlling behaviour. Additionally, the Swedish NBW as well as previous research, highlights the importance of healthcare providers and social service providers informing the police in cases with a potential risk of IPH perpetration [6, 24].

## Sociodemographic characteristics related to health care contacts during the year prior to the homicide

When perpetrators who had had primary care and specialized outpatient care contacts in the year prior to index were compared to their matched controls with healthcare contacts, a larger proportion of perpetrators were unemployed and received social benefits. In fact, the majority of perpetrators who sought healthcare in the year prior to the offence were unemployed, whereas the opposite was true for the control group. The finding concurs with previous literature in that socioeconomic disadvantage, including unemployment, has been identified as an important determinant associated with perpetration of IPH [8, 10, 11, 20, 51]. A possible explanation for the findings in our study is that most unemployed IPH perpetrators may have a range of different mental and physical problems. Previous research suggests that perpetrators with a range of mental health symptoms and diagnoses are also more likely to be physically ill and therefore, they may be more likely to contact healthcare services [52]. Due to the small sample size of perpetrators, we were not able to analyse differences in sociodemographic characteristics between perpetrators and controls with mental or behavioural disorders diagnosed

in primary care and/or outpatient care settings. Future studies should explore whether differences in mental and behavioural disorders between IPH perpetrators and matched controls from the general population also exist when it comes to sociodemographic factors such as employment status and social benefits.

It is important to bear in mind that despite significant healthcare utilization among IPH perpetrators in this study, the extent of healthcare utilization and the prevalence of mental and behavioural disorders may have been underestimated. Previous research suggests that many IPH perpetrators never seek help from any healthcare service [22, 52]. One earlier study from England and Wales, showed that approximately one-third of IPH perpetrators who suffered from mental illness never sought any mental health services [22].

## Methodological considerations

One strength of this study is the inclusion of matched controls from the general population, allowing for comparison between IPH perpetrators and individuals in the general population. This in turn contributes to more knowledge about how IPH perpetrators differ from the general population in terms of healthcare utilization and mental and behavioural disorders. Another strength is the use of data based on high-quality national and regional registries, maintained by the government and with mandatory registration of diagnoses [3, 31]. Since this study compared mental and behavioural disorders in perpetrators and matched controls from a general perspective, specific clinical diagnoses in the ICD-10 code range F00-F99, where not extracted and investigated.

The major limitation of this study is the small perpetrator sample size, specifically concerning female perpetrators. This may have influenced the statistical power, and consequently the possibility to correctly identify any difference between perpetrators and controls. These findings must thus be interpreted with caution. Moreover, the small sample size did not allow for additional multivariable analyses.

## Conclusions

Overall, this study shows that male IPH perpetrators had significantly more registered healthcare contacts and mental and behavioural disorders in the month and year prior to the homicide (including the day of the homicide), compared to their matched controls. Female IPH perpetrators had more often received specialized outpatient care and inpatient care in the year prior to the homicide, compared to controls. This study has also shown that a larger proportion of IPH perpetrators who were in contact with primary care and specialized outpatient care in the year prior to the offence were unemployed and had received social benefits. The findings of this study reveal that IPH perpetrators frequently contact healthcare services shortly before the homicide and that they may seek care for a wide range of health issues, including mental health problems. Concurring with previous research, our findings suggest that healthcare services may be the last point of contact for many perpetrators before they kill their partner [16]. It is thus important that healthcare professionals gain necessary skills for routinely enquiring about IPV, with the aim of identifying individuals at risk of perpetrating IPH. Furthermore, healthcare professionals must have adequate time and opportunity to listen actively and explore the intimate relationship history of patients at risk of perpetration, as well as to enquire about their mental, physical and emotional needs as this is critical for implementation of appropriate support and referral [16]. Thus far, there has been a considerable lack of attention devoted to IPV and IPH perpetrators. In order to prevent IPH, it is important that routine enquiry addresses not only the victims of IPV, but also the perpetrators of IPV.

## Acknowledgments

The authors would like to thank Västra Götaland Region Competence Centre on Intimate Partner Violence (VKV) and the Regional Police Authority in Gothenburg who helped to make this research possible.

## Author Contributions

**Conceptualization:** Solveig Lövestad, Karin Örmon, Viveka Enander, Gunilla Krantz.

**Data curation:** Gunilla Krantz.

**Formal analysis:** Solveig Lövestad.

**Methodology:** Solveig Lövestad.

**Project administration:** Karin Örmon, Viveka Enander.

**Writing – original draft:** Solveig Lövestad.

**Writing – review & editing:** Karin Örmon, Viveka Enander, Gunilla Krantz.

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
