## [Decision Letter · Decision Letter 0]

5 Oct 2023

PONE-D-23-04399Healthcare utilization, mental disorders and behavioural disorders among perpetrators of intimate partner homicide in 2010-2016: a registry-based case-control study from SwedenPLOS ONE

Dear Dr. Lövestad,

Thank you for submitting your manuscript to PLOS ONE. After careful consideration, we feel that it has merit but does not fully meet PLOS ONE’s publication criteria as it currently stands. Therefore, we invite you to submit a revised version of the manuscript that addresses the points raised during the review process.

Dear authors, I kindly request that you take into consideration all the comments from the reviewers. I believe they are relevant and appropriate.

We look forward to receiving your revised manuscript.

Kind regards,

José J. López-Goñi

Academic Editor

PLOS ONE

Additional Editor Comments:

Dear authors, I kindly request that you take into consideration all the comments from the reviewers. I believe they are relevant and appropriate.

Reviewers' comments:

Reviewer's Responses to Questions

**Comments to the Author**

1. Is the manuscript technically sound, and do the data support the conclusions?

Reviewer #1: Partly

Reviewer #2: Yes

2. Has the statistical analysis been performed appropriately and rigorously? 

Reviewer #1: I Don't Know

Reviewer #2: I Don't Know

3. Have the authors made all data underlying the findings in their manuscript fully available?

Reviewer #1: Yes

Reviewer #2: No

4. Is the manuscript presented in an intelligible fashion and written in standard English?

Reviewer #1: Yes

Reviewer #2: Yes

5. Review Comments to the Author

Reviewer #1: Thank you for the opportunity to review this important work. The manuscript is well-written and compelling. However, it would benefit from some minor adjustments, as follows;

Abstract: The first two sentences are quite repetitive, I would consider revising.

Methods: There is no information about how the sample was matched, it would be good to know how respondents were chosen and the methods that were used to match them to the perpetrators. It is also important to include any power analysis that helped to calculate the sample size for the study since the sample is so small.

Reviewer #2: Thank you for the opportunity to review the paper entitled “Healthcare utilization, mental disorders and behavioural disorders among perpetrators of intimate partner homicide in 2010-2016: a registry-based case-control study from Sweden”.

This is an original research with novel results, for which I congratulate the authors. Having said this, I consider that some aspects should be revised and improved so that the article can be publishable.

1) Introduction:

- The differential explanation between IPH committed by men and by women should have a somewhat broader theoretical substrate focused on the aim of the study.

- The introduction is mainly focused on mental health problems, could be said something more specific about the contact with the health care system in order to justify the aims of the study?

- In relation to the above, why so much theoretical background on the differences between IPH perpetrators and non-lethal IPV or other non-partner victims (focused on mental health problems) to conclude that it is important to study the differences between IPH perpetrators and the general population? Why is this difference important and for what?

2) Methods:

- Given the wide variety of disorders included in the ICD-10 classification, it would be interesting to know which are the specific diagnoses (at least the most frequent), as well as the number of diagnoses of each participant and the number of times they use health care. This would lead to a clearer identification of the differences with the general population in order to make a proposal for a practical application of the results.

- In the same way, when the health care contact is for reasons other than mental-health reasons, it would be advisable to identify those reasons.

- It would be necessary to clearly define what is meant by “social benefits”. Are they only economic benefits or other types of services?

3) Discussion and conclusions:

- The results are really interesting as they support the evidence about a higher use of health care resources of IPH, although the percentage is low. But I consider that the reflection on their implications and their practical application in the healthcare field should be much more profound.

o Mental health diagnoses and physical health diagnoses would be comparable?

o What can be the real role of healthcare professionals in the detection of potential homicides?

o How are they going to identify or screen them?

o Have those professionals previously identified IPV in order to identify a risk of homicide?

o What would be the ethical and legal implications of this?

o Would it be advisable to being coordinated with other agents, such as police, justice, social services…

6. PLOS authors have the option to publish the peer review history of their article (what does this mean?). If published, this will include your full peer review and any attached files.

Reviewer #1: **Yes: **Flora Cohen

Reviewer #2: No

---

## [Author Response · Author response to Decision Letter 0]

27 Dec 2023

Response to reviewer # 1

Reviewer’s comments: 

Thank you for the opportunity to review this important work. The manuscript is well-written and compelling. However, it would benefit from some minor adjustments, as follows;

Abstract: The first two sentences are quite repetitive, I would consider revising.

Our comments:

Thank you very much for your valuable comments and suggestions for improvement of this paper. We have made changes in the two first sentences in the abstract according to the reviewer’s suggestion: “Little is known about intimate partner homicide (IPH) perpetrator´s healthcare contacts and mental health problems before the killing. The aim was to compare male and female IPH perpetrators with matched controls from the general population by analysing differences in healthcare utilization and mental and behavioural disorders”.

We hope this has improved the text, and made the first two sentences less repetitive.

Reviewer’s comments: 

Methods: There is no information about how the sample was matched, it would be good to know how respondents were chosen and the methods that were used to match them to the perpetrators. It is also important to include any power analysis that helped to calculate the sample size for the study since the sample is so small.

Our comments:

In the methods section, we explain how the controls were matched, i.e. for each offender, 10 controls were randomly selected and matched for sex, birth year and residential area at the time of index. In addition to this, we have now added information that the random selection of the controls was performed by Statistics Sweden which is a governmental agency, and that this approach has been used in several other registry-based studies previously conducted in Sweden. (line 207-209 in the Revised Manuscript, methods section). We have also added examples of sources from previous registry-based studies performed in Sweden that have used this method previously. 

Study sample size (power) was not calculated a-priori since the number of male and female IPH perpetrators in the Västra Götaland Region between 2000-2016 was fixed and not possible to modify. The same number of matched controls has been used in previous registry-based studies performed in Sweden; examples of these studies are referred to in the manuscript (line 209).

Response to reviewer # 2

Reviewer’s comments:

Thank you for the opportunity to review the paper entitled “Healthcare utilization, mental disorders and behavioural disorders among perpetrators of intimate partner homicide in 2010-2016: a registry-based case-control study from Sweden”.

This is an original research with novel results, for which I congratulate the authors. Having said this, I consider that some aspects should be revised and improved so that the article can be publishable.

Our response

Thank you very much for your valuable comments and suggestions for improvement of this paper. We are grateful for these inputs and we hope that we have managed to clarify any doubts or unanswered questions with our responses, comments and changes in this paper. Our answers are found below, after each reviewer comment.

Reviewer’s comments:

1) Introduction:

- The differential explanation between IPH committed by men and by women should have a somewhat broader theoretical substrate focused on the aim of the study.

Our comments

We have added more theoretical substrate according to your suggestions, explaining the differences between male and female perpetrated IPH in the introduction section, line 75-76, line 92-101 and line 101-110 in the revised manuscript. Further we have added more theory about female IPH and mental health problems in the introduction section, line 118-124. 

Reviewer’s comments:

- The introduction is mainly focused on mental health problems, could be said something more specific about the contact with the health care system in order to justify the aims of the study?

Our comments

We have added more information about contact with the healthcare system under the introduction section, line 90-110. Most of this information overlaps with theory provided for the previous point above, i.e. male perpetrators may contact healthcare services, primarily for other reasons than mental health issues whereas female perpetrators may contact healthcare services for being exposed to IPV. Since both male and female perpetrators may contact healthcare services for a range of health issues other than mental health problems, we believe that it is important to examine general healthcare utilization of IPH perpetrators.

Reviewer’s comments:

- In relation to the above, why so much theoretical background on the differences between IPH perpetrators and non-lethal IPV or other non-partner victims (focused on mental health problems) to conclude that it is important to study the differences between IPH perpetrators and the general population? Why is this difference important and for what?

Our comments

We wanted to provide a clear outline of the findings from previous research that compares perpetrators of IPH with perpetrators of IPV and other types of homicides. This, since we wanted to highlight the differences and similarities of previous results based on comparisons between IPH perpetrators and other types of perpetrators without simplifying too much. In addition, and to our knowledge, only one previous study has compared IPH perpetrators with controls from the general population (this is mentioned in the introduction) and therefore we could not add much of previous findings based on such a comparison. Social and mental health issues may be overrepresented among homicide perpetrators and when perpetrators of IPH are compared with other types of perpetrators, they seem to be more conventional in many aspects. We wanted to add new and more knowledge to the picture by examining the differences between perpetrators and the general population without any known history of IPH. In line with this, we have added more information in the introduction section, line 159-165.

Reviewer’s comments:

2) Methods:

- Given the wide variety of disorders included in the ICD-10 classification, it would be interesting to know which are the specific diagnoses (at least the most frequent), as well as the number of diagnoses of each participant and the number of times they use health care. This would lead to a clearer identification of the differences with the general population in order to make a proposal for a practical application of the results.

Our comments:

From previous research performed by Belfrage & Rying (2004), Lysell et al. (2016) and Caman et al. (2016), we already got valuable knowledge about specific psychiatric diagnosis among IPH perpetrators in Sweden. We wanted to add new knowledge by investigating whether IPH perpetrators in the Region of Västra Götaland, had more registered psychiatric diagnoses, regardless of specific psychiatric diagnoses, compared to individuals from the general population. We have added information in the introduction section (line 126-130) where we describe that we want to study mental and behavioural disorders from a broader perspective. If we would have focused on specific diagnoses only, we may have missed valuable information about potential differences between IPH perpetrators and the general population, as well as information about perpetrator’s general mental health diagnoses.

Reviewer’s comments:

- In the same way, when the health care contact is for reasons other than mental-health reasons, it would be advisable to identify those reasons.

Our comments:

In line with our answer in the previous point we wanted to examine health care utilization from a broader perspective. This since IPH perpetrators may seek care for a range of different reasons other than mental health problems. Our aim was to examine whether IPH perpetrators and individuals from the general population differed in healthcare contacts regardless of diagnosis. One reason for this is that a majority of IPH perpetrators are men and previous research indicates that societal norms may lead men to seek care primarily for reasons other than specific mental health concerns We have added information about this in the introduction, line 92-101. 

Reviewer’s comments: 

It would be necessary to clearly define what is meant by “social benefits”. Are they only economic benefits or other types of services?

Our comments

We have clarified what is included in “social benefits”(methods section, line 273-274).

Reviewer’s comments: 

3) Discussion and conclusions:

- The results are really interesting as they support the evidence about a higher use of health care resources of IPH, although the percentage is low. But I consider that the reflection on their implications and their practical application in the healthcare field should be much more profound.

o Mental health diagnoses and physical health diagnoses would be comparable?

Our comments:

We may have misunderstood the question, but we do not believe that mental health and physical health diagnoses are comparable, and our aim has not been to compare mental health diagnoses with physical disorders. We hope that our previous answers and comments have clarified any ambiguities or misunderstandings. If not, we will gladly clarify further doubts. 

Reviewer’s comments: 

o What can be the real role of healthcare professionals in the detection of potential homicides?

o How are they going to identify or screen them?

o Have those professionals previously identified IPV in order to identify a risk of homicide?

Our comments:

From our perspective (which also is the recommendation from the National Board on Health and Welfare), healthcare providers need to inquire patients about IPV in order to prevent IPH. Please see or response (added text) in the revised manuscript under the discussion section, line 405-409. 

Reviewer’s comments: 

o What would be the ethical and legal implications of this?

Our comments:

Questions about IPV perpetration needs to be asked in a non-judgemental way and involving follow-up visits, clear communication with the patient about potential implications of the disclosure and clear routines and procedures in case of disclosure. This is added under the discussions section, 421-435. From an ethical perspective where lives may be in danger; If healthcare providers suspect hat perpetrators may be dangerous for a current or former partner, the recommendation is that healthcare providers inform the police. This is also added in the discussion section (line 434-435). 

Reviewer’s comments: 

o Would it be advisable to being coordinated with other agents, such as police, justice, social services…

Our comments:

Yes, we have added information about this (line 431-435).

---

## [Decision Letter · Decision Letter 1]

12 Jan 2024

PONE-D-23-04399R1Health care utilization, mental disorders and behavioural disorders among perpetrators of intimate partner homicide in 2000-2016: a registry-based case-control study from SwedenPLOS ONE

Dear Dr. Lövestad,

Thank you for submitting your manuscript to PLOS ONE. After careful consideration, we feel that it has merit but does not fully meet PLOS ONE’s publication criteria as it currently stands. Therefore, we invite you to submit a revised version of the manuscript that addresses the points raised during the review process.

Dear authors, on this occasion, one of the reviewers was unable to assess the changes made. Based on my own judgement, I believe that the manuscript has been substantially improved. There is only one final contribution left from reviewer 2. Please evaluate this.

Congratulations on the work you have done.

We look forward to receiving your revised manuscript.

Kind regards,

José J. López-Goñi

Academic Editor

PLOS ONE

Journal Requirements:

Additional Editor Comments:

Dear authors, on this occasion, one of the reviewers was unable to assess the changes made. Based on my own judgement, I believe that the manuscript has been substantially improved. There is only one final contribution left from reviewer 2. Please evaluate this.

Congratulations on the work you have done.

Reviewers' comments:

Reviewer's Responses to Questions

**Comments to the Author**

1. If the authors have adequately addressed your comments raised in a previous round of review and you feel that this manuscript is now acceptable for publication, you may indicate that here to bypass the “Comments to the Author” section, enter your conflict of interest statement in the “Confidential to Editor” section, and submit your "Accept" recommendation.

Reviewer #2: All comments have been addressed

2. Is the manuscript technically sound, and do the data support the conclusions?

Reviewer #2: (No Response)

3. Has the statistical analysis been performed appropriately and rigorously? 

Reviewer #2: (No Response)

4. Have the authors made all data underlying the findings in their manuscript fully available?

Reviewer #2: (No Response)

5. Is the manuscript presented in an intelligible fashion and written in standard English?

Reviewer #2: (No Response)

6. Review Comments to the Author

Reviewer #2: I reiterate my congratulations to the authors for their work. The responses to the comments are very pertinent and the manuscript has improved.

I would like to make one last suggestion for change before the document is finally accepted:

In the Discussion, line 446, reads "...concerns for the health and well-being of both the perpetrator and others are expressed [47]". I think that this "others" should make visible and expressly refer (in the way the authors consider most appropriate) to the people who are affected by this violence, especially the direct victims.

7. PLOS authors have the option to publish the peer review history of their article (what does this mean?). If published, this will include your full peer review and any attached files.

Reviewer #2: No

---

## [Author Response · Author response to Decision Letter 1]

27 Jan 2024

Journal Requirements:

Our comments:

We have checked our reference list and ensured that it is complete and correct; there are no retracted papers included. In the previous revision and after suggestions from reviewers, we added some clarifications and more theory. Thus, the references that we added to the previous revision (submitted in December 2023) are the following references:

• Voce I, Bricknell S. Female perpetrated intimate partner homicide: Indigenous and non-Indigenous offenders: Australian Institute of Criminology; 2020.

• Vickery A. Men's Help-Seeking for Distress: Navigating Varied Pathways and Practices. Frontiers in sociology. 2021;6:724843-. doi: 10.3389/fsoc.2021.724843.

• Bonomi AE, Anderson ML, Rivara FP, Thompson RS. Health Care Utilization and Costs Associated with Physical and Nonphysical-Only Intimate Partner Violence. Health services research. 2009;44(3):1052-67. doi: 10.1111/j.1475-6773.2009.00955.x.

• Petersen R, Gazmararian J, Andersen Clark K. Partner violence: implications for health and community settings. Women's health issues. 2001;11(2):116-25. doi: 10.1016/S1049-3867(00)00093-1.

• Shilan C, Joakim S, Katarina H. Mental Disorders and Intimate Partner Femicide: Clinical Characteristics in Perpetrators of Intimate Partner Femicide and Male-to-Male Homicide. Frontiers in psychiatry. 2022;13. doi: 10.3389/fpsyt.2022.844807.

• Weizmann-Henelius G, Matti Grönroos L, Putkonen H, Eronen M, Lindberg N, Häkkänen-Nyholm H. Gender-Specific Risk Factors for Intimate Partner Homicide: A Nationwide Register-Based Study. Journal of interpersonal violence. 2012;27(8):1519-39. doi: 10.1177/0886260511425793.

• Ludvigsson J, Almqvist C, Bonamy A-K, Ljung R, Michaëlsson K, Neovius M, et al. Registers of the Swedish total population and their use in medical research. European Journal of Epidemiology. 2016;31(2):125-36. doi: 10.1007/s10654-016-0117-y

.

• Fazel S, Lichtenstein P, Grann M, Långström N. Risk of violent crime in individuals with epilepsy and traumatic brain injury: a 35-year Swedish population study. PLoS medicine. 2011;8(12):e1001150-e. doi: 10.1371/journal.pmed.1001150.

• Lysell H, Runeson B, Lichtenstein P, Långström N. Risk factors for filicide and homicide: 36-year national matched cohort study. The journal of clinical psychiatry. 2014;75(2):127-32. doi: 10.4088/JCP.13m08372.

• Salinas Fredricson A, Naimi-Akbar A, Adami J, Lund B, Rosén A, Hedenberg-Magnusson B, et al. Diseases of the musculoskeletal system and connective tissue in relation to temporomandibular disorders—A SWEREG-TMD nationwide case-control study. PloS one. 2022;17(10):e0275930-e. doi: 10.1371/journal.pone.0275930.

• Lysell H, Dahlin M, Viktorin A, Ljungberg E, D'Onofrio BM, Dickman P, et al. Maternal suicide - Register based study of all suicides occurring after delivery in Sweden 1974-2009. PloS one. 2018;13(1):e0190133-e. doi: 10.1371/journal.pone.0190133.

• Michau LMA, Horn JM, Bank ABA, Dutt MJD, Zimmerman CP. Prevention of violence against women and girls: lessons from practice. The Lancet (British edition). 2015;385(9978):1672-84. doi: 10.1016/S0140-6736(14)61797-9.

• Gerlock AA, Grimesey JL, Pisciotta AK, Harel O. Documentation of screening for perpetration of intimate partner violence in male veterans with PTSD. The American journal of nursing. 2011;111(11):26. doi: 10.1097/01.NAJ.0000407296.10524.d7.

• Kimberg LS. Addressing Intimate Partner Violence with Male Patients: A Review and Introduction of Pilot Guidelines. Journal of general internal medicine : JGIM. 2008;23(12):2071-8. doi: 10.1007/s11606-008-0755-1.

• Portnoy GA, Colon R, Gross GM, Adams LJ, Bastian LA, Iverson KM. Patient and provider barriers, facilitators, and implementation preferences of intimate partner violence perpetration screening. BMC health services research. 2020;20(1):746-. doi: 10.1186/s12913-020-05595-7.

• Taket A, Nurse J, Smith K, Watson J, Shakespeare J, Lavis V, et al. Routinely asking women about domestic violence in health settings. BMJ. 2003;327(7416):673-6. doi: 10.1136/bmj.327.7416.673.

• The National Board of Health and Welfare. Våld i nära relationer- Handbok för socialtjänsten , hälso och sjukvården och tandvården (Domestic violence – A handbook for social services, health care and dental care). 2023.

Additional Editor Comments:

Dear authors, on this occasion, one of the reviewers was unable to assess the changes made. Based on my own judgement, I believe that the manuscript has been substantially improved. There is only one final contribution left from reviewer 2. Please evaluate this.

Congratulations on the work you have done.

Our comments:

Thank you for this positive comment and news. We have made suggestions according to the reviewer’s comment/suggestion. Please see our answer below.

Reviewer #2: 

I reiterate my congratulations to the authors for their work. The responses to the comments are very pertinent and the manuscript has improved. I would like to make one last suggestion for change before the document is finally accepted:

In the Discussion, line 446, reads "...concerns for the health and well-being of both the perpetrator and others are expressed [47]". I think that this "others" should make visible and expressly refer (in the way the authors consider most appropriate) to the people who are affected by this violence, especially the direct victims.

Our comment:

First of all, we are extremely grateful for all the help we have received in improving the content of this manuscript. According to your suggestion, in line 426-428 (in the revised manuscript) we have changed/added the text as follows: “Furthermore, it is important to address IPV as a healthcare issue, expressing concerns for the health and well-being of both the perpetrator, the victim, and their children.” We hope that this has clarified the content of the sentence.

---

## [Editor Report · Decision Letter 2]

30 Jan 2024

Health care utilization, mental disorders and behavioural disorders among perpetrators of intimate partner homicide in 2000-2016: a registry-based case-control study from Sweden

PONE-D-23-04399R2

Dear Dr. Lövestad,

We’re pleased to inform you that your manuscript has been judged scientifically suitable for publication and will be formally accepted for publication once it meets all outstanding technical requirements.

Kind regards,

José J. López-Goñi

Academic Editor

PLOS ONE
---

## [Editor Report · Acceptance letter]

13 Feb 2024

PONE-D-23-04399R2 

PLOS ONE

Dear Dr. Lövestad, 

I'm pleased to inform you that your manuscript has been deemed suitable for publication in PLOS ONE. Congratulations! Your manuscript is now being handed over to our production team.

Kind regards, 

on behalf of

Dr. José J. López-Goñi 

Academic Editor

PLOS ONE